# Removal of Excess Alkali from Sodium Naphthenate Solution by Electrodialysis Using Bilayer Membranes for Subsequent Conversion to Naphthenic Acids

**DOI:** 10.3390/membranes11120980

**Published:** 2021-12-14

**Authors:** Aslan Achoh, Ilya Petriev, Stanislav Melnikov

**Affiliations:** 1Physical Chemistry Department, Faculty of Chemistry and High Technologies, Kuban State University, 350040 Krasnodar, Russia; achoh-aslan@mail.ru; 2Department of Physics, Kuban State University, 350040 Krasnodar, Russia; petriev_iliya@mail.ru; 3Laboratory of Problems of Stable Isotope Spreading in Living Systems, Southern Scientific Centre of the RAS, 344000 Rostov-on-Don, Russia; 4Department of Physics, K.G. Razumovsky Moscow State University of Technologies and Management (The First Cossack University), 109004 Moscow, Russia

**Keywords:** electrodialysis, ion-exchange membrane, naphthenic acids conductivity, bilayer membranes, current efficiency

## Abstract

The processing of solutions containing sodium salts of naphthenic acids (sodium naphthenate) is in high demand due to the high value of the latter. Such solutions usually include an excessive amount of alkali and a pH of around 13. Bipolar electrodialysis can convert sodium naphthenates into naphthenic acids; however, until pH 6.5, the naphthenic acids are not released from the solution. The primary process leading to a decrease in pH is the removal of excess alkali that implies that some part of electricity is wasted. In this work, we propose a technique for the surface modification of anion-exchange membranes with sulfonated polyetheretherketone, with the formation of bilayer membranes that are resistant to poisoning by the naphthenate anions. We investigated the electrochemical properties of the obtained membranes and their efficiency in a laboratory electrodialyzer. Modified membranes have better electrical conductivity, a high current efficiency for hydroxyl ions, and a low tendency to poisoning than the commercial membrane MA-41. We propose that the primary current carrier is the hydroxyl ion in both electromembrane systems with the MA-41 and MA-41M membranes. At the same time, for the modified MA-41M membrane, the concentration of hydroxyl ions in the anion-exchanger phase is higher than in the MA-41 membrane, which leads to almost five-fold higher values of the specific permeability coefficient. The MA-41M membranes are resistant to poisoning by naphthenic acids anions during at least six cycles of processing of the sodium naphthenate solution.

## 1. Introduction

Disposal of refinery by-products is mainly random and depends on favorable local conditions, particularly the presence of a nearby chemical plant. For example, in most cases, sulfuric alkalis are not used and are dumped into water bodies. They are one of the most harmful types of wastewater pollution by oil refineries [1].

At present, petroleum acids are still the only class of petroleum oxygen-containing compounds that have found wide application in various fields of industry [2]. Among this class of acids, naphthenic acids occupy a special place. Some areas of application of naphthenic acids include: solvents for polymers, dyes, and rubber; varnish components, antiseptics; additives for printing inks; and anti-knock additives for motor fuels [3,4]. It should be noted that the limiting factor in expanding the area of application of naphthenic acids is the limited nature of their reserves, reduction in the production of oils rich in these acids, and lack of cheap technologies for their processing. The need for naphthenic acids is many times higher than their possible production [5].

On an industrial scale, naphthenic acids are extracted from petroleum distillates (mainly diesel and kerosene fractions) by alkaline extraction with an aqueous solution of sodium hydroxide. In this case, sodium salts of naphthenic acids form, which move into the aqueous phase (in what follows, such a solution with a complex chemical composition will be called sodium naphthenate). The treatment of sodium naphthenate solution with a strong acid (usually sulfuric) leads to the formation of a two-phase system, the upper layer of which (organic phase) contains naphthenic acids, and the lower (aqueous)—sodium sulfate [2]. The formation of large volumes of acidic sodium sulfate solutions makes the technology incompatible with “green” chemistry principles.

Electrodialysis is used in several industrial processes associated with the removal of ions from aqueous solutions [6,7,8], concentration and separation of substances [9], electromembrane synthesis [10], and processing solutions in the agro-food industry [11,12]. In recent decades, the electrodialysis technology for the recovery of salts, including salts of organic acids, into the corresponding acids and alkalis (bases) has been developed [13]. The demand for membrane technologies is primarily due to the development of areas of “green chemistry” and, in particular, using biomass fermentation products as a source of organic acids [6,14,15,16].

Electrodialysis of sodium naphthenate allows us not only to obtain a finished product—naphthenic acids—but also to return the alkali to the production of sodium naphthenate. The studies carried out earlier showed that in the pH range from 8 to 7, a mixture of naphthenic acids and sodium naphthenates is formed—the so-called acidol. This product has a significantly higher viscosity and electrical resistance [17] than sodium naphthenate solution, making the electrodialysis process implementation in this pH range impractical. It is advisable to divide the process of obtaining naphthenic acids into two stages: extraction of excess alkali lowering the pH value from 13 to 8 and conversion of naphthenic acids in the pH range 6.5–2.

Several negative phenomena arise during the recovery of acids and alkalis from aqueous-organic solutions of salts, primarily caused by organic components. The main effect is the poisoning of membranes with organic substances, which increases their electrical resistance, a decrease in the limiting current, and selectivity [18]; another factor is the change in the transport-structural characteristics of ion-exchange membranes [19,20]. A way out of this problem is the surface modification of the ion-exchange membrane. A highly selective layer on the membrane surface eliminates the contact of the membrane volume with poisoning substances, thereby maintaining high electrical conductivity and other transport characteristics.

Several physicochemical approaches are known for forming selective layers on the surface of membranes [21,22]. To separate ions with different hydrophilic properties and ionic radiuses, the sieving effect and hydrophilization or hydrophobization of the membrane surface are used [20].

In an earlier work, we proposed to use a three-chamber unit cell in the membrane stack of a bipolar electrodialyzer to prevent contact of the organic phase with the anion-exchange layer of the bipolar membrane [13]. The use of bilayer membranes, obtained by applying a layer of a cation-exchanger on an anion-exchange membrane substrate, will make it possible to switch to two-chamber unit cells, reducing capital and operating costs and accelerating the processing of sodium naphthenate. We define “bilayer membranes” as a special type of ion-exchange membrane consisting of two layers, at least one of which should possess ion-exchange properties. We specifically refer to them as “bilayer membranes” to emphasize that their main function is not predetermined at the stage of synthesis but can change depending on operational conditions [23].

The possibility of using an electrodialyzer with bilayer membranes will make it possible to extract excess alkali from sodium naphthenate solution, excluding membrane poisoning with naphthenic acids anions.

## 2. Materials and Methods

### 2.1. Objects of the Study

The objects of the study were the Russian heterogeneous ion-exchange membranes MK-40 and MA-41 (JSC “Shchekinoazot”, Shchekino, Russia) and bilayer membranes MA-41M, with a thin selective layer of sulfonated polyetheretherketone.

Membranes MK-40 and MA-41 are commercially available. They are produced by hot pressing a mixture consisting of a fine powder of an ion-exchanger and polyethylene taken in a volume ratio of 2:1. By the type of ionogenic groups, MK-40 is a strongly acidic sulfonic cation-exchanger; MA-41 is a strongly basic quaternary ammonium anion-exchanger.

The physicochemical characteristics of the studied membranes are shown in Table 1.

For surface modification of anion-exchange membranes, sulfonated polyetheretherketone was obtained. Sulfonation of polyether ketone (PEEK) was carried out according to the following procedure [24]: 10 g of granulated PEEK was placed in 500 mL of concentrated sulfuric acid, and then left for 120 h at room temperature with vigorous stirring (the PEEK granules were completely dissolved in sulfuric acid). After this time, the resulting solution was slowly poured into 1500 mL of ice water. The resulting polymer precipitated, filtered off under a vacuum, and washed with distilled water until the filtrate was neutral. The filtered polymer was dried in a vacuum desiccator.

The resulting polymer was dissolved in dimethylformamide. As a result, a 10% solution of sulfonated polyetheretherketone (SPEEK) was obtained.

To obtain a MA-41M membrane, a thin layer of sulfonated polyetheretherketone was applied to the surface of the MA-41 membranes. The liquid sulfonated polyetheretherketone (SPEEK) was applied to the prepared part of the ion-exchange membrane at the rate of 0.4 mL/dm^2^ and left to dry for 24 h to remove the excess solvent (DMF). Before starting the application of the SPEEK solution, the membrane surface was degreased with carbon tetrachloride. After drying, a 50% solution of acetic acid was applied to the membrane surface at the rate of 0.2 mL per 1 dm^2^. The resulting membrane had a modifying film thickness of 10 ± 1 μm. The thicknesses of the modifying films and membranes were determined using an Absolute Digimatic MDH Mitutoyo electronic micrometer, with an error of 0.5 μm. The thickness of the studied membranes was measured in a swollen state.

The membranes were equilibrated with a 0.1 M sodium hydroxide solution in which they were stored until the moment of the study.

### 2.2. Electrodialysis Tests

We determined the efficiency of modified MA-41M membranes in comparative tests in the membrane stack of the laboratory electrodialyzers with a two-chamber unit cell (Figure 1). In the first case, the electrodialyzer consisted of alternating MK-40 cation-exchange membranes and an MA-41 anion-exchange membrane. In the second test, the membrane stack was composed of the MK-40 cation-exchange membranes and the surface-modified MA-41M membranes.

The alkali removal process was studied in a galvanostatic mode at a current density of 1 A/dm^2^. A model solution of sodium naphthenate with a concentration of 1.3 M was fed into the desalination chambers, a 0.1 M sodium hydroxide solution circulated in the concentration chambers, and the electrode chambers were washed sequentially with 0.1 M sodium hydroxide. Electrodialysis was carried out in the range of pH values 13 to 8. The characteristics of the electrodialyzer are presented in Table 2.

The chambers were filled with a mesh separator-turbulizer to prevent sagging of the membranes due to the pressure difference in the chambers and an increase in the limiting electrodiffusion current on the membranes due to a decrease in the thickness of the diffusion layers near the membranes. The free volume of each chamber, not occupied by the separator-turbulizer, was 80% of the chamber volume.

The concentration and desalination chamber flow was kept constant and equal to 30 L/h through the electrode chambers—sufficient to remove the gases formed during operation.

### 2.3. Conductivity Measurement

The mercury contact method was used to study the electrical conductivity of membranes [25]. A mercury contact cell with a membrane was connected to a PARSTAT 4000 impedance meter according to a two-electrode circuit to obtain the resistance value. The frequency range in which the measurements were carried out ranged from 500 kHz to 10 Hz, with a zero constant current component and an alternating current signal amplitude of 100 μA. The frequency range in which the measurements were carried out ranged from 500 kHz to 10 Hz, with a zero constant current component and an alternating current signal amplitude of 100 μA.

Extrapolation of the linear part of the spectrum in the mid-frequency region allows one to obtain the value of the membrane’s active (ohmic) resistance. The resulting value is converted into electrical conductivity according to the equation:(1)$$\kappa_{m}^{AC}=\frac{l}{RS}$$
where $\kappa_{m}^{AC}$ is a specific electrical conductivity on alternating current, *S*/*m*; *l* is membrane thickness, *m*; *R* is membrane electrical resistance, Ohm; and *S* is sample area, m^2^.

### 2.4. Current–Voltage Curve Measurement

The current–voltage characteristics were investigated using a rotating membrane disk with a surface equally accessible in diffusion and electrical respects [26]. The current–voltage characteristics of the membranes were recorded in a galvanostatic mode, with a stepwise increase in the current density. The rotation speed of the membrane disk was varied from 100 to 500 rpm and was measured using an optical-mechanical transducer with a built-in digital unit. A mixed solution of 0.1 M NaOH and 0.1 M NaNF was used as a model solution. The feed rate of the solution into the cathode chamber was 7.5 ± 0.1 mL/min. The concentration of anions was determined using the method of potentiometric titration.

The effective (Hittorf) ion transport numbers (*T_j_*) were determined by the formula:(2)$$T_{j}=\frac{\left(c_{j}-c_{j}^{0}\right)VF}{I}$$
where $c_{j}$ is the electrolyte concentration at the outlet of the cathode chamber, M; $c_{j}^{0}$ is the concentration of the feed solution, M; *V* is the solution volumetric flow, L/s; *F* is the Faradays constant, C/mol; and *I* is the applied current, A.

Based on the dependences of the effective transport numbers of ions, the specific permeability coefficients were calculated:(3)$$P_{1,2}=\frac{T_{1}c_{2}^{0}}{T_{2}c_{1}^{0}}$$
where $T_{1}$ and $T_{2}$ are the effective transport numbers of hydroxyl and naphthenate ions, respectively; and $c_{1}^{0}\;$ and $c_{2}^{0}$ concentration of hydroxyl and naphthenate ions, respectively, in the bulk of the solution, M.

The theoretical value of the limiting electrodiffusion current was calculated using the Peers equation and the diffusion coefficients of electrolytes in solution:(4)$$i_{lim}=\frac{2FDc}{\delta }$$
where c is the concentration of electrolytes NaOH or NaNf in the bulk solution, M; δ is the diffusion boundary layer thickness, m; and D is the diffusion coefficients of electrolytes NaOH or NaNf in the bulk solution at infinite dilution, m^2^/s.

## 3. Results and Discussion

### 3.1. Membrane Conductivity

To determine the resistance to the poisoning of the modifying film in solutions of naphthenic acids, we measured the electrical conductivity of sulfonated polyetheretherketone (SPEEK) in the form of a film with a thickness of 200 μm and the initial membrane MA-41. The conductivity of the membranes was measured after equilibration with mixed solutions of naphthenic acids and alkali. The total concentration of NaOH and NaNF remained constant and equal to 0.2 M, while the fraction of naphthenate ions in the solution varied from 0 to 100%. The results of studying the electrical conductivity of SPEEK films and MA-41 membranes are shown in Figure 2.

Figure 2 shows that the electrical conductivity of the MA-41 membrane tends to be zero at the molar fraction of naphthenate ions in the membrane equal to 0.4. From the point of view of the percolation theory [27], this implies that a system of hydrated ion pairs fixed ion-counterion (a percolation cluster) is ruptured. From a practical point of view, this would result in a shallow current passing through the system, limiting the electrodialysis process’s application.

In the case of the SPEEK film, the electrical conductivity practically does not change because the fixed groups in the modifying film have a negative charge. In this case, naphthenic acid anions are co-ions, and their sorption is blocked due to electrostatic repulsion. Based on these data, we concluded that SPEEK is stable in alkaline solutions containing naphthenic acid anions.

### 3.2. Investigation of the Electromembrane Process of Removing Alkali from Sodium Naphthenate by Electrodialysis

#### 3.2.1. Current–Voltage Characteristics of Membranes

For the effective operation of the electrodialyzer, it is required that the value of the limiting current be as large as possible. It is known [28] that when applying a modifying layer with antipolar ionic groups with ionic groups of the substrate membrane, a significant decrease in the limiting electrodiffusion current is observed, sometimes reaching ten or more times a drop.

Figure 3 shows the I–V characteristics of the modified MA-41M ion-exchange membrane and the initial MA-41 membrane obtained on a setup with a rotating membrane disk in a mixed solution containing 0.1 M sodium naphthenate and 0.1 M sodium hydroxide (the concentration of sodium ions in the solution is 0.2 M).

Deposition of a cation-exchange film on the surface of the MA-41 membrane leads to a decrease in the limiting current, but under experimental conditions, this decrease is only two times. Such a slight difference can be explained by the thickness of the cation-exchange film (10 μm), which effectively traps naphthenic acid ions but does not prevent the transfer of hydroxyl ions. Hydroxyl ions are co-ions, with respect to the modifying layer of sulfonated polyetheretherketone. Their concentration inside the coating is much lower than the exchange capacity. Their concentration at the modified membrane’s cation exchanger/anion exchanger interface rapidly decreases with increasing current density. Such a change in the external diffusion kinetics to the internal diffusion kinetics for bilayer and bipolar membranes is described in the literature [29,30].

Using the Peers equation, we calculated the limiting current values for a system with pure electrolytes—sodium hydroxide and sodium naphthenate (Figure 3, lines 3 and 4). The experimental and theoretical values of the limiting currents are given in Table 3. The limiting current for the MA-41 membrane practically coincides with the limiting current for a pure 0.1 M NaOH solution. Since the experimental data were obtained in a mixed solution with a total concentration of OH^−^ and Nf^−^ ions of 0.2 M, it can be assumed that there is practically no transport of naphthenate ions in the studied electromembrane system; the current flows due to the transport of hydroxyl ions. We can also assume that OH^−^ ions are the primary carriers in the membrane volume for both MA-41 and the modified MA-41M membrane.

It is important to note that the deposition of a cation-exchange layer on an anion-exchange membrane substrate leads to a so-called bipolar boundary. There is a possibility of the water-splitting reaction in the system at high current densities (the overlimiting current mode) [33]. The current density of 1 A/dm^2^ selected for the research is much lower than the limiting current density; from our previous studies of bilayer membranes [23], we know that in the underlimitting current mode, the main charge carriers in the system will be “salt” ions (in the studied case, hydroxyl ions present in the solution). The chosen current density limits the possibility of water splitting on the bilayer membranes, thus preventing chemical reactions involving hydrogen ions in the treated solution.

#### 3.2.2. Specific Selectivity of Anion-Exchange Membranes

To determine the efficiency of the surface modification of the anion-exchange membrane, we determined the specific selectivity for hydroxyl ions. A mixed solution of 0.05 M sodium naphthenate and 0.1 M sodium hydroxide was used as a model solution. The results of studies of selective permeability obtained on a laboratory electrodialyzer are shown in Figure 4.

The data obtained ensures that both membranes are characterized by a predominant selectivity to the hydroxyl ion. For an unmodified MA-41 membrane at the beginning of the experiment, the selective permeability coefficient is approximately 10. According to [34], the specific permeability coefficient is determined by the ratio of the diffusion coefficients and the concentrations of competing ions in the membrane. From the sorption isotherm given in [17] it follows that the concentration of naphthenate ions in the membrane phase should be lower than the concentration of hydroxyl ions. At the same time, the diffusion coefficient of the naphthenate ion in the solution should be at least an order of magnitude lower than that of the hydroxyl ion [32]. The difference in the diffusion coefficients should be even more significant in the membrane phase, which is associated with the relay mechanism of the transfer of the hydroxyl ion. Thus, the observed value of the selective permeability coefficient, as in the case of the current–voltage characteristic, indicates that the main carrier in the membrane phase is the hydroxyl ion.

The surface modification of the MA-41 membrane with a SPEEK cation-exchange film leads to a more than five-fold increase in the specific permeability for hydroxyl ions compared to the initial MA-41 anion-exchange membrane. This effect can be explained by the fact that, under the above experimental conditions, naphthenate anions cannot penetrate the cation-exchange film and further into the volume of the anion-exchange membrane due to both electrostatic repulsion and the specific interaction of naphthenate anions and the surface of the SPEEK film. The reasons for the specific repulsion of naphthenate anions can be size effects (the naphthenate anion is larger than the size of the ion channel) or hydrophilic/hydrophobic interactions of anions with the film material.

The decrease in the specific permeability coefficient over time is associated with a decrease in the concentration of hydroxyl ions in the external solution and, consequently, an increase in the proportion of naphthenate ions in the anion exchanger.

#### 3.2.3. Two-Chamber Electrodialyzer Tests

An example of the change in the pH value of the solution in the desalting chamber during the electrodialysis process is shown in Figure 5.

As can be seen, when using a two-chamber cell with a MA-41 membrane, there is practically no change in pH over time, which indicates the absence of the transfer of hydroxyl ions from the desalting chamber. As in the study of electrical conductivity, the probable reason is the blocking of ion channels by naphthenic acid anions. As a result, mobile hydroxyl ions can be transported across the membrane only by diffusion through the intergel solution. Although the MA-41 anion-exchange membrane is heterogeneous and the fraction of the intergel solution can reach 0.2–0.3, this process proceeds at a low rate, which does not allow for efficient alkali extraction from sodium naphthenate.

The data obtained for the modified membrane, on the other hand, indicate a high permeability for hydroxyl ions as a result of which it is possible to reach a value of pH = 8 and to extract, if necessary, an excess alkali from the naphthenate solution.

The conclusions drawn are confirmed by calculating the hydroxyl ion current efficiencies for the studied membranes (Figure 6).

Figure 6 shows that the hydroxyl ion current efficiency on the initial MA-41 membrane is initially lower than for the modified membrane. The difference between the values of the transfer number at the initial points (at high pH values) can be explained by the fact that the poisoning of the MA-41 membrane occurs quickly during the first few minutes of contact with the sodium soap solution. Over time, the fraction of sorbed naphthenate ions in the MA-41 membrane increases, which leads to a further decrease in the current efficiency.

For the MA-41M membrane, the current efficiency for hydroxyl ions is practically independent of the pH value and averages 0.75 over the entire investigated range.

#### 3.2.4. Stability of the Modified Membranes

Long-term stability tests were carried out to determine the applicability of modified ion-exchange membranes during prolonged contact with naphthenic acid anions, processing six portions of sodium naphthenate on the same membranes in a membrane stack. The dependence of the resistance of the unit cell on the time of electrodialysis is shown in Figure 7.

As can be seen from Figure 7, the value of the resistance of the unit cell, in the case of MA-41 anion-exchange membranes, increases to 30 Ohm dm already in the first cycle. In this regard, further tests of the membrane package with the MA-41 anion-exchange membrane were not carried out.

In the case of modified membranes, the maximum resistance value is about 7 Ohm·dm when the pH of the solution reaches 7.5. We should note that the increase in the resistance of the unit cell for the case with the MA-41M membrane from 2 Ohm to 7 Ohm is due to the desalting of the solution (removal of excess alkali). Six independent experiments were carried out on the same membranes, which showed the stability of the obtained surface modified membrane MA-41M.

Studies have shown that it is possible to extract excess alkali and reduce pH from 13 to 8 using a two-chamber electrodialyzer with a modified MA-41 anion-exchange membrane, excluding the poisoning of anion-exchange membranes.

## 4. Conclusions

Studies have shown that when processing sodium naphthenate, electrodialysis removal of excess alkali and a decrease in pH to 7.5–8 using surface-modified anion-exchange membranes is possible. The deposition of a cation-exchange film on the surface of the MA-41 anion-exchange membrane prevents “poisoning” of the membrane with naphthenate anions, apparently due to the electrostatic and specific repulsion of the latter by the surface of the modifying film. In this case, there was a decrease in the limiting electrodiffusion current by approximately two times compared to the initial membrane-substrate, which could be explained by the transition of the external diffusion kinetics of ion transfer to the inside diffusion kinetics.

Two-layer membranes MA-41M possess sufficient stability in a solution of soap, which is confirmed by the invariability of the properties of the electromembrane package during several cycles of removing excess alkali from the sodium naphthenate solution. In this case, the current efficiency for hydroxyl ions remained at a high level and averaged 0.75 in each measurement cycle.

## Figures and Tables

**Figure 1 membranes-11-00980-f001:**
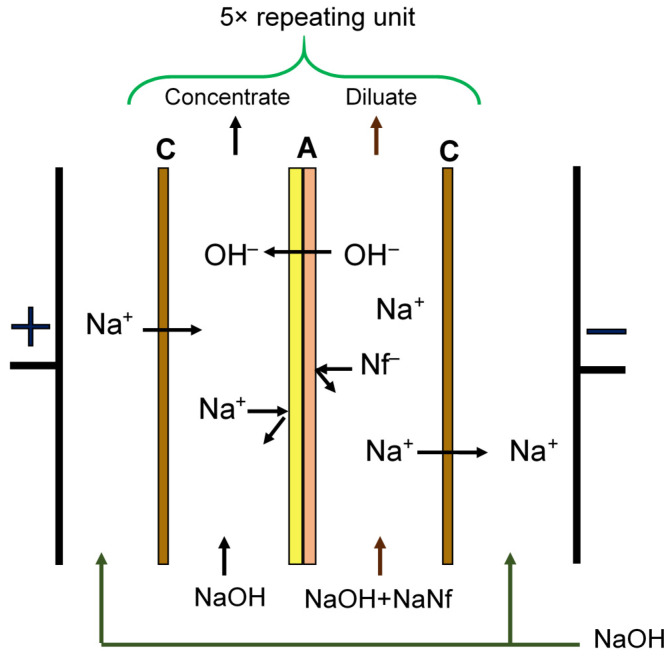
Scheme of a unit cell of a laboratory electrodialyzer: C—cation-exchange membranes MK-40, A—anion-exchange membranes MA-41 or MA-41M.

**Figure 2 membranes-11-00980-f002:**
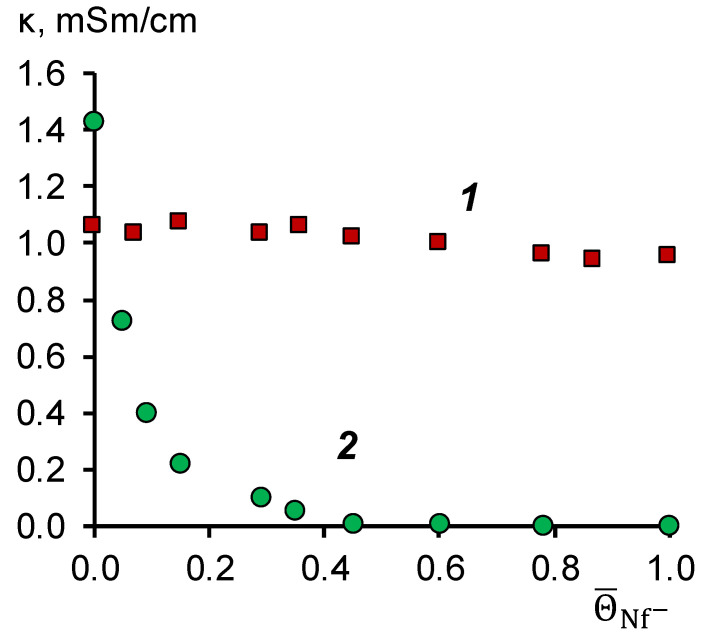
Dependence of the electrical conductivity of the SPEEK film (1) and the MA-41 anion-exchange membrane (2) on the fraction of naphthenate anions in a mixed solution of NaOH + NaNf, with a total concentration of 0.2 M.

**Figure 3 membranes-11-00980-f003:**
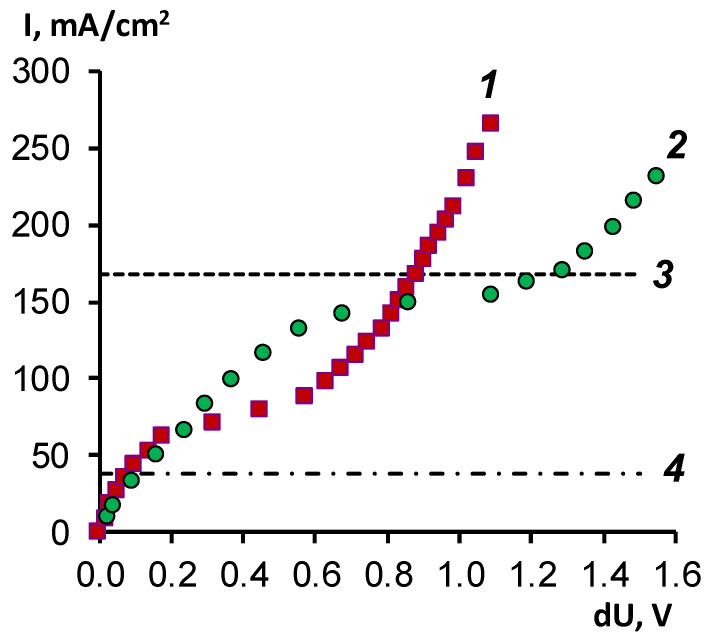
General current–voltage characteristics of the modified MA-41M membrane (1) and the initial MA-41 membrane (2) at a membrane disk rotation speed of 100 rpm, (3) calculation of the limiting current according to Equation (4) for 0.1 M NaOH, and (4) calculation of the limiting current according to Equation (4) for 0.1 M NaNf.

**Figure 4 membranes-11-00980-f004:**
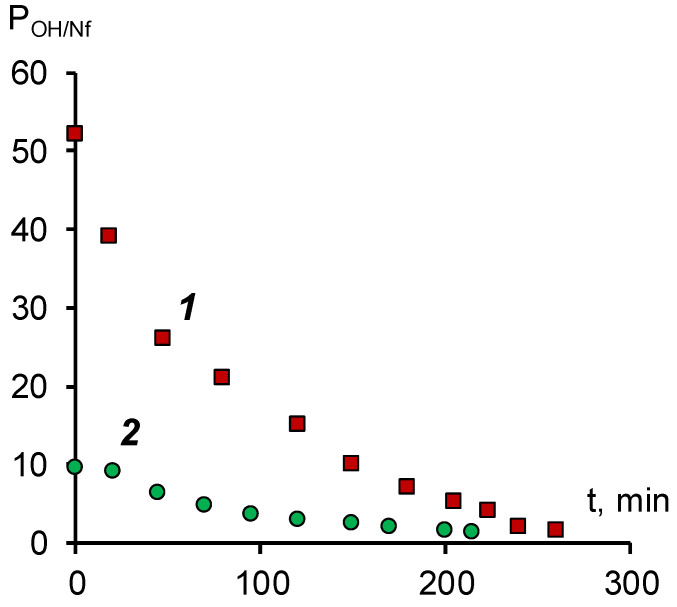
Dependence of the coefficient of selective permeability for the hydroxyl ion/naphthenate ion pair during electrodialysis in a galvanostatic mode with a current density of 0.2 A/dm^2^. 1—modified membrane MA-41M, 2—membrane MA-41.

**Figure 5 membranes-11-00980-f005:**
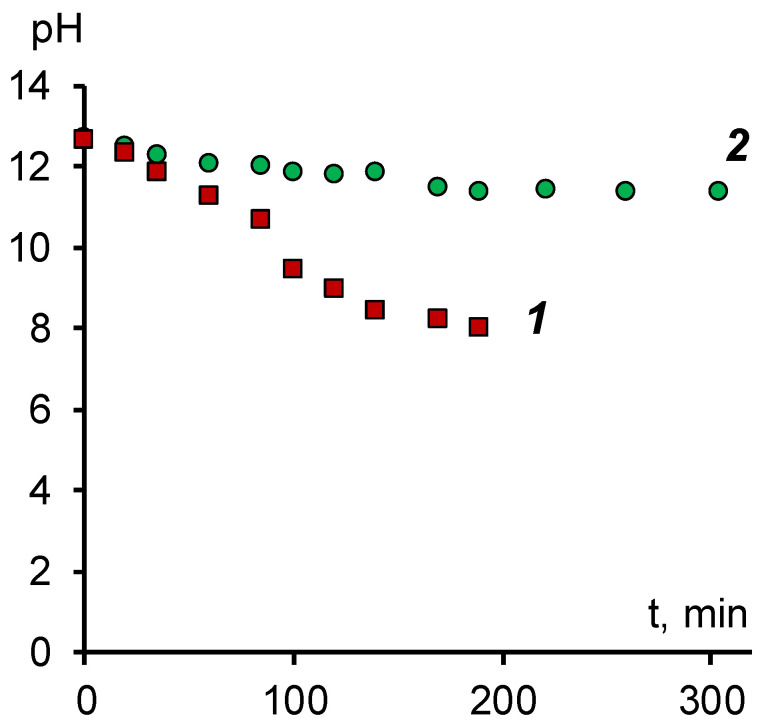
Dependence of the pH value of the sodium naphthenate solution on the operating time of the electrodialyzer. 1—modified membrane MA-41M, 2—membrane MA-41.

**Figure 6 membranes-11-00980-f006:**
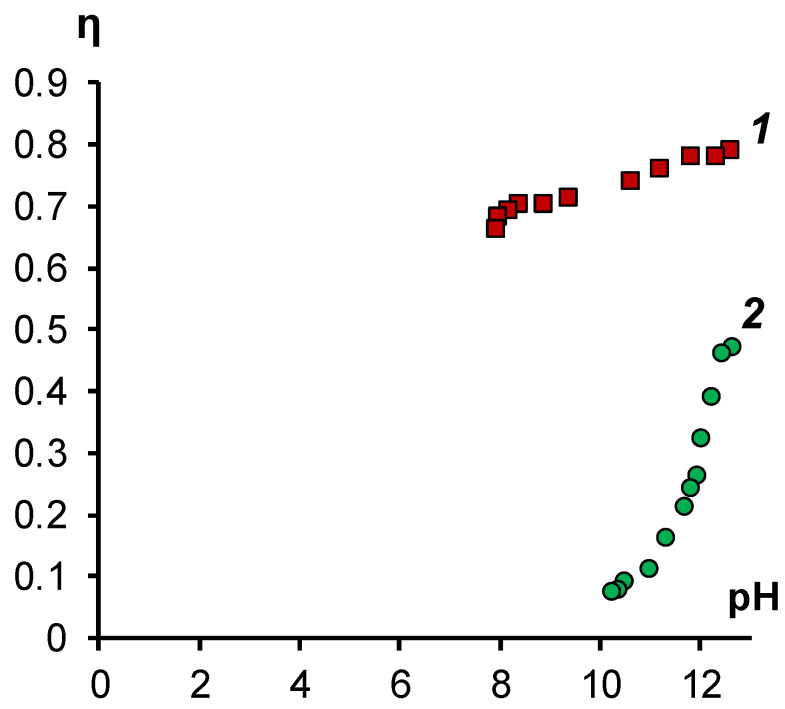
Dependence of the current efficiency for hydroxyl ions on the pH value of sodium naphthenate solution in the process of electrodialysis in galvanostatic mode with a current density of 1 A/dm^2^. 1—modified membrane MA-41M, 2—membrane MA-41.

**Figure 7 membranes-11-00980-f007:**
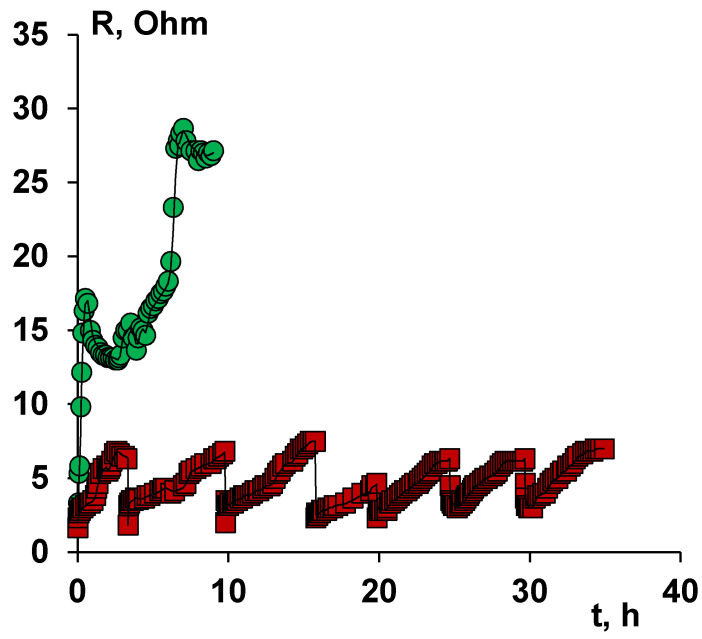
Dependence of the resistance of the unit cell on the time of electrodialysis. Electrodialysis was carried out in a galvanostatic mode with a current density of 1 A/dm^2^.

**Table 1 membranes-11-00980-t001:** Physicochemical characteristics of the studied ion-exchange membranes.

Membrane	MK-40	MA-41
Fixed groups	-SO^3−^	-N+(CH_3_)_3_
Ion-exchange resin	KU-2-8	AV-17-8
Ion-exchange capacity, mmol/g-wet	1.08	0.91
Water uptake, %	33	36
Swollen membrane thickness, microns	540	530

**Table 2 membranes-11-00980-t002:** Characteristics of laboratory electrodialyzer.

Elementary Cell	Two-Chamber
Number of elementary cells	5 pcs
Solutions flow mode	Parallel from bottom up
Membranes	
anion-exchange	MA-41, MA-41M
cation-exchange	MK-40
Materials	
anode	Ruthenium oxide titanium anode
cathode	Stainless steel
spacers	polyethylene
Channel dimensions	
length	200 mm
width	50 mm
height	0.9 m

**Table 3 membranes-11-00980-t003:** Diffusion coefficients and limiting current in the electromembrane system.

	Electrolyte Diffusion Coefficients, ×10^−9^ m^2^/s	Limiting Current Value, mA/cm^2^
0.1 M NaOH solution	2.13 [31]	168 ^1^
0.1 M NaNf solution	0.11 [32]	39 ^1^
MA-41 in the mixed solution	–	135 ± 3 ^2^
MA-41M in the mixed solution	–	72 ± 3 ^2^

^1^ calculated using Peers equation, ^2^ experimental limiting current.

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
