# Peer review of "Removal of Excess Alkali from Sodium Naphthenate Solution by Electrodialysis Using Bilayer Membranes for Subsequent Conversion to Naphthenic Acids"

_membranes, 2021, doi:10.3390/membranes11120980_

Round 1

Reviewer 1 Report

The submitted maqnuscript explain the application of the modified anion exchange membrane that avoid poisening with naphtenate anion during the electrodialysis process. The organization, writing and presentation are quite good and the experiments support the conclusion. I only noticed minor typos 

Figure 5 line 223 must be changed to Figure 2, or this sentence should be rearranged: 

only through diffusion through the intergel solution (line 330)

Otherwise I would recommend the publication of the submitted article in the current form. 

Best regards

Author Response

Reviewer #1

The submitted manuscript explains the application of the modified anion exchange membrane that avoids poisoning with naphthenate anion during the electrodialysis process. The organization, writing, and presentation are quite good and the experiments support the conclusion. I only noticed minor typos

Answer: We highly appreciate the reviewers` opinion of our work. Below we will give detailed answers to the comments.

>>Figure 5 line 223 must be changed to Figure 2, or this sentence should be rearranged;

Answer: We changed the number of the figure in the sentence. Now it reads: “Figure 2 shows that the electrical conductivity…”

>>only through diffusion through the intergel solution (line 330)

Answer: The sentence was rewritten as: “As a result, mobile hydroxyl ions can be transported across the membrane only by diffusion through the intergel solution.”

Otherwise, I would recommend the publication of the submitted article in its current form.

Best regards

Reviewer 2 Report

The authors provide a ED technology to remove alkali from sodium naphthenate, this work provide sufficient details, so it is recommend to publish after small revision.

  1. What is the difference of bipolar membrane and bilayer membrane? Also, the monovalent ion exchange membrane? The author should make this clear.
  2. Fig.2, so the membrane conductivity reached 0, so how the system continute operation?
  3. Is there any calculation to identify the selectivity of the bilayer membrane?
  4. Is there any OH- leakage and water splitting phenomenon during the process? what does the effect on this system?

Author Response

Reviewer #2

The authors provide an ED technology to remove alkali from sodium naphthenate, this work provides sufficient details, so it is recommended to publish after a small revision.

Answer: We highly appreciate the reviewers` opinion of our work. Below we will give detailed answers to the comments.

>>1. What is the difference of bipolar membrane and bilayer membrane? Also, the monovalent ion exchange membrane? The author should make this clear.

Answer: We define “bilayer membranes” as a special type of ion-exchange membranes consisting of two layers at least one of which should possess ion-exchange properties. In our works, we usually deal with bilayer membranes in which one of the layers (modifying layer) is 10 times thinner than the other (membrane-substrate). From this perspective, we can call bipolar membranes bilayer membranes. But, we specifically refer to them as “bilayer membranes” to emphasize that their main function is not predetermined at the stage of synthesis but can change depending on operational conditions. The transport of salt ions and water-splitting products through a bilayer membrane is complex and is affected by the current density, nature, and concentration of the solution, composition of the modifying layer as we have shown in [Melnikov S. et al. Water Splitting and Transport of Ions in Electromembrane System with Bilayer Ion-Exchange Membrane // Membranes (Basel). 2020. Vol. 10, â„– 11. P. 346.].

We added the following text to the last paragraph of the introduction (starting from line 90): “We define “bilayer membranes” as a special type of ion-exchange membranes consisting of two layers at least one of which should possess ion-exchange properties. We specifically refer to them as “bilayer membranes” to emphasize that their main function is not pre-determined at the stage of synthesis but can change depending on operational conditions [23].”

>>2. Fig.2, so the membrane conductivity reached 0, so how the system continute operation?

Answer: The obtained data for electrical conductivity of the MA-41 membrane shows that when the fraction of naphthenate ions in the membrane will be higher than 0,4 the current passing through it will also tend to zero. What we see from this data is in agreement with data provided in figure 5 we see almost no transport of ions in an ED module with MA-41 membranes.

We added the following text starting at line 226: “From a practical point of view, this would result in shallow current passing through the system, limiting the electrodialysis process's application.”

>>3. Is there any calculation to identify the selectivity of the bilayer membrane?

Answer: To answer this question we must define the selectivity term. If the reviewer is asking for selectivity between co-ions and counter ions (the most usual definition of selectivity regarding ion-exchange membranes) then there is no such definition for the membranes used in our study. Because we use cation-exchanger as modifying layer and anion-exchanger as membrane-substrate, the co-ions in one of the layers will be counterions in the other. Because of this, the concentration of salt on the interface cation-exchanger/anion-exchanger will be different from ion-exchanger/solution. This uncertainty and the rise of Donnan potential on the inner interface make it difficult to come with an expression for counterion/co-ion selectivity of the bilayer membrane. Kedem and Katchalsky gave one of the possible solutions for a series-layered membrane for transport numbers of co-ions [Kedem O., Katchalsky A. Permeability of composite membranes. Part 3.—Series array of elements // Trans. Faraday Soc. 1963. Vol. 59, â„– 19. P. 1941–1953.].

If we define the selectivity as a specific selectivity between various types of counterions (with regard to the membrane-substrate) then it can be calculated from experimental data using Eqs. (2) and (3) in the manuscript. If we consider a more general case then theoretical analysis was given in [Achoh A.R. et al. Electrochemical Properties and Selectivity of Bilayer Ion-Exchange Membranes in Ternary Solutions of Strong Electrolytes // Membr. Membr. Technol. 2021. Vol. 3, â„– 1. P. 52–71.; Zabolotsky V.I. et al. Permselectivity of bilayered ion-exchange membranes in ternary electrolyte // J. Memb. Sci., 2020. Vol. 608, P. 118152.].

>>4. Is there any OH- leakage and water splitting phenomenon during the process? what does the effect on this system?

Answer: There is a possibility for water-splitting in the current system due to the nature of the bilayer membrane. But we chose to work at relatively low current density (see the CVC in figure 3) which excludes the water-splitting. The OH- leakage from the concentration chamber is also highly improbable because the concentration (0.1 M) is lower than in the naphthenate solution (0.3 M) and the migration transport also opposes the transport of OH through anion-exchange or bilayer membrane.

To better show this we modified the text in section 3.2.1 (starting from line 279) in the following way: “It is important to note that the deposition of a cation-exchange layer on an anion-exchange membrane-substrate leads to a so-called bipolar boundary. There is a possibility of the water-splitting reaction in the system at high current densities (the overlimiting current mode) [33]. The current density of 1 A/dm2 selected for the research is much lower than the limiting current density; from our previous studies of bilayer membranes [23], we know that in the underlimitting current mode, the main charge carriers in the system will be “salt” ions (in the studied case hydroxyl ions present in the solution). The chosen current density limits the possibility of water-splitting on the bilayer membranes, thus preventing chemical reactions involving hydrogen ions in the treated solution.”